# Release Kinetics and In Vitro Characterization of Sodium Percarbonate and Calcium Peroxide to Oxygenate Bioprinted Tissue Models

**DOI:** 10.3390/ijms23126842

**Published:** 2022-06-20

**Authors:** Dongxu Ke, Carlos Kengla, Sang Jin Lee, James J. Yoo, Xuesong Zhu, Sean Vincent Murphy

**Affiliations:** 1Wake Forest Institute for Regenerative Medicine, Wake Forest University School of Medicine, Winston-Salem, NC 27101, USA; dongxu.ke@wsu.edu (D.K.); kenglacv@wfu.edu (C.K.); sjlee@wakehealth.edu (S.J.L.); jyoo@wakehealth.edu (J.J.Y.); 2Department of Orthopedics, First Affiliated Hospital of Soochow University, Suzhou 215006, China

**Keywords:** sodium percarbonate, calcium peroxide, oxygen release, cell viability, cell differentiation, bioprinting

## Abstract

Oxygen-generating materials have been used in several tissue engineering applications; however, their application as in situ oxygen supply within bioprinted constructs has not been deeply studied. In this study, two oxygen-generating materials, sodium percarbonate (SPO) and calcium peroxide (CPO), were studied for their oxygen release kinetics under a 0.1% O_2_ condition. In addition, a novel cell-culture-insert setup was used to evaluate the effects of SPO and CPO on the viability of skeletal muscle cells under the same hypoxic condition. Results showed that SPO had a burst oxygen release, while CPO had a more stable oxygen release than SPO. Both SPO and CPO reduced cell viability when used alone. The addition of catalase in SPO and CPO increased the oxygen release rate, as well as improving the viability of skeletal muscle cells; however, CPO still showed cytotoxicity with catalase. Additionally, the utilization of 1 mg/mL SPO and 20 U catalase in a hydrogel for bioprinting significantly enhanced the cell viability under the hypoxic condition. Moreover, bioprinted muscle constructs could further differentiate into elongated myotubes when transferring back to the normoxic condition. This work provides an excellent in vitro model to test oxygen-generating materials and further discover their applications in bioprinting, where they represent promising avenues to overcome the challenge of oxygen shortage in bioprinted constructs before their complete vascularization.

## 1. Introduction

Sodium percarbonate (SPO) and calcium peroxide (CPO) are two common oxygen-generating materials. SPO is an adduct of hydrogen peroxide and sodium carbonate. The oxygen is produced by the decomposition of hydrogen peroxide into oxygen and water. This reaction is fast because it is accelerated by the alkaline environment due to sodium carbonate [1]. CPO has peroxide chemically bound to the calcium ion, resulting in a gradual decomposition to hydrogen peroxide and then to oxygen and water [2,3]. Hence, the oxygen release rate of CPO is much slower than that of SPO.

SPO and CPO are promising materials for various tissue-engineering applications. They can be used to prevent tissue necrosis after severe injuries and to improve wound healing and tissue regeneration [1,3,4,5]. They are also promising when incorporated into bioinks to provide oxygen for improved cell viability during the vascularization process in large bioprinted tissue constructs; however, studies on this topic have not yet been published [6]. One concern for oxygen-generating materials is their cytotoxicity, caused by the accumulation of hydrogen peroxide during the reaction. It has been reported that the accumulation of hydrogen peroxide can result in superoxide anion and hydroxyl radicals, leading to cell death [7,8,9]. One solution to prevent the accumulation of hydrogen peroxide is the addition of catalase, which is an enzyme that can accelerate the decomposition of hydrogen peroxide [3,6,10].

Studies using SPO or CPO as oxygen depots have been published for various applications [1,3,4,5,11,12]. Harrison et al. reported a novel poly(D,L-lactide-co-glycolide) (PLGA) film including the presence of SPO for treating ischemic tissues [1]. Results showed that the film could produce oxygen for 24 h under normal conditions and avoid necrosis of ischemic tissue in a mouse model for 3 days. In another study, CPO was added to an antioxidant polyurethane polymeric material to prepare a cryogel for tissue engineering applications [4]. Oxygen could still be detected after 10 days under 1% O_2_ hypoxic conditions. In addition, the cryogel improved in vitro cell viability for 5 days and prevented tissue necrosis of ischemic skin flaps in a dorsal skin flap model for 9 days. Even though there are a few studies using SPO and CPO as the oxygen supply for wound-healing applications, there are no studies comparing the release kinetics of SPO and CPO in an extreme hypoxic condition. An appropriate in vitro model to establish an initial screening for oxygen-generating materials, such as SPO and CPO, is still lacking but would be crucial to expanding their use to various tissue-engineering applications, especially bioprinting.

In this study, the release kinetics of SPO and CPO with and without catalase was studied in a 0.1% O_2_ condition. Then, skeletal muscle cells were used to study the effects of SPO and CPO with and without catalase on in vitro cell viability using a novel cell-culture-insert system, which can be useful for testing any oxygen generation system in various tissue-engineering applications. The viability of skeletal muscle cells in the presence of oxygen-generation materials was also characterized through live–dead staining and an MTT assay. Finally, one composition was selected to be incorporated with a hydrogel to prepare bioprinted muscle constructs, followed by in vitro characterizations using live–dead staining. In addition, the differentiation of skeletal muscle cells in bioprinted constructs was further evaluated by immunocytochemical staining when transferred from hypoxic to normoxic conditions.

## 2. Results

### 2.1. Oxygen Release

For the oxygen release from SPO and CPO, oxygen concentration peaks were detected 1 h after initiating the release study, as shown in Figure 1a. The oxygen release rate of 1 mg/mL SPO was much higher than the other compositions. The oxygen concentration during release from 1 mg/mL SPO substantially decreased from 31.17 ± 1.87 mg/mL to 6.48 ± 0.39 mg/mL in the initial 47 h and then gradually decreased to 3.14 ± 0.19 mg/mL after 12 days of release study. The oxygen released from 1 mg/mL CPO gradually decreased from 6.02 ± 0.36 mg/mL to 4.63 ± 0.28 mg/mL after 10 days and then substantially decreased to 0.11 ± 0.01 mg/mL at day 11 of the release study. The oxygen release from 0.1 mg/mL SPO and 0.1 mg/mL CPO substantially decreased after the peak, and their oxygen concentrations were maintained till day 2 and day 7, respectively. With the presence of catalase in SPO and CPO, their oxygen release rate became much faster, allowing for a higher peak oxygen concentration, as shown in Figure 1b. After the peak, the oxygen levels in all compositions decreased at a higher rate, such that SPO and CPO approached depletion after one day of the release study.

### 2.2. Cell Viability and Proliferation

Cells with only SPO and CPO showed significantly decreased live cell areas and MTT values; however, samples with SPO/catalase and CPO/catalase had similar or improved cell viability and proliferation compared to the hypoxic control at day 1. Among all compositions, 1 mg/mL SPO with 20 U catalase resulted in significantly enhanced live cells and higher MTT levels than the hypoxic control at day 1, which was similar to the cellular condition of the normoxic control, as shown in Figure 2b–d.

Moreover, samples with 1 mg/mL SPO/20 U catalase and 1 mg/mL CPO/20 U catalase were further tested for extended time points in vitro. As shown in Figure 3a–c, significantly increased numbers of live cells were observed along with significantly increased MTT levels in the 1 mg/mL SPO/20 U catalase samples at days 7 and 12. However, samples cultured with 1 mg/mL CPO/20 U catalase showed cytotoxic effects and detached from the well plate, while demonstrated significantly lower cell viability in the same assays. Both samples were assessed in comparison with the hypoxic control.

### 2.3. In Vitro Characterizations of Bioprinted Muscle Construct

A design with a dimension of 10 × 10 × 0.5 mm was used to prepare bioprinted muscle constructs, as shown in Figure 4a. Green indicates PCL, red indicates bioink 1, and blue indicates bioink 2. The fibrinogen-based hydrogel and fibrinogen-based hydrogel incorporated with the 1 mg/mL SPO/20 U catalase showed excellent printability, as shown from the image of the post-printing muscle construct in Figure 4b. Both compositions maintained high cell viability after culturing under hypoxic conditions for 7 days; however, the hydrogel incorporated with the 1 mg/mL SPO/20 U catalase (94.2 ± 3.8%) demonstrated significantly higher cell viability than the control (84.5 ± 5.1%), as shown in Figure 4c.

Samples with hydrogel inserts were further tested for their differentiation ability after transferal from hypoxic conditions to normoxic conditions. The differentiation capacity was assessed by myosin staining of differentiated myotubes. Following 15 days under hypoxic conditions, cells in the hydrogel with 1 mg/mL SPO/20 U catalase stained positive for myotubes after being transferred to normoxic conditions and continued exhibiting proliferation and differentiation for 8 days; however, cells in the hydrogel hypoxic control showed weak evidence of well-differentiated myotubes, as shown in Figure 5a. In addition, the number of myotubes and the elongated myotube percentage from the hydrogel with 1 mg/mL SPO/20 U catalase were significantly higher than the control, as shown in Figure 5b–c.

## 3. Discussion

Oxygen-generating materials have been used in different tissue engineering applications; however, the optimum oxygen release rates required for these applications differ. For example, wound-healing applications require a fast and relatively short oxygen release to prevent tissue necrosis after severe injuries; however, for engineered tissues or organs, a long-term and stable oxygen release is necessary to maintain cell viability and tissue functionality before the vascularization process is complete [13,14,15]. In our release studies of SPO and CPO, we observed that SPO had a burst oxygen release, while CPO had a more stable oxygen release than SPO, which was expected due to the different chemical decomposition kinetics of SPO and CPO [1,3]. In addition, 1 mg/mL of SPO and 1 mg/mL CPO could maintain the oxygen level close to the saturation level under hypoxic conditions for 10 days, as shown in Figure 1a. With the presence of catalase, the oxygen-producing reactions of SPO and CPO became much faster, showing an undersaturated oxygen level after one day, as shown in Figure 1b. This correlated well with our knowledge that the presence of catalase can accelerate the decomposition of hydrogen peroxide, resulting in faster oxygen release [16].

Besides oxygen release, the biocompatibility of oxygen-generating materials is also crucial to their application. In this study, a novel cell-culture-insert setup was used to evaluate the biocompatibility of SPO and CPO, as shown in Figure 1a. Adding cell medium with SPO and CPO above the filter of the cell culture inserts meant that unreacted SPO and CPO could not attach to the bottom of the well plates to interfere with the cellular activities. In addition, cells could absorb the oxygen released from the SPO and CPO through medium diffusion. This novel cell-culture-insert setup was an excellent tool for testing SPO and CPO and could also be useful for testing other types of oxygen-generating materials.

In this study, in the presence of SPO and CPO, cytotoxicity was evident after one day of culture. Significantly reduced live cell areas and MTT levels indicated inferior cell viability in experimental groups compared to the hypoxic control. This discovery corresponded well with results from previous studies indicating that the accumulation of hydrogen peroxide can reduce cell viability [6]. The presence of catalase remarkably improved cell viability by accelerating the decomposition of hydrogen peroxide, as shown in Figure 2b–d; however, CPO remained cytotoxic in the presence of catalase, which was not found in previous studies [3,4,5]. In previous studies, CPO was combined with other biomaterials to prepare scaffolds and followed by characterizations in vitro or in vivo that indicated few cytotoxic effects [4,5,11,17]. However, in this study, cells had a direct response to CPO, which should provide a more accurate assessment for the cytotoxicity of CPO during its reaction. One possible reason for the cytotoxicity of CPO concerns its byproduct, calcium hydroxide, which has been reported to be cytotoxic in previous studies [18,19].The CPO used in this study was only 70% pure (which was the highest purity on the market), and the impurities in the CPO might have contributed to the cytotoxicity. Additionally, the cytotoxic effects of CPO should be closely dependent on its dose and testing environment. Previous results might not be useful to predict the conclusion in this study.

A total of 1 mg/mL SPO or CPO with 20 U catalase was further evaluated to study their long-term biocompatibility. Using the same culture insert setup, 1 mg/mL SPO/20 U catalase showed significantly higher cell viability in live–dead images and a higher MTT level than the 1 mg/mL CPO/20 U catalase and the hypoxic control after 7 and 12 days of culture, as shown in Figure 3a–c. In addition, this result was superior to those reported in previous approaches to maintaining cell viability under hypoxic conditions [4,20]. The reason might have been due to the effects of the oxygen vibrating environment on cellular proliferation. It has been reported that changes in oxygen tension and the appropriate culturing time under hypoxic conditions may enhance the proliferation of C2C12 cells [21]. In this study, the addition of 1 mg/mL SPO/20 U catalase resulted in oxygen environmental changes for 2 days, as shown in Figure 1, which should have been more beneficial for the proliferation of C2C12 cells compared to the hypoxic control after one day, as shown in Figure 2. Moreover, it has been reported that a temporary hypoxic condition can improve the proliferation of C2C12 cells [21]. Hence, the stable hypoxic condition did not cause cell death immediately on day 1 but at the prolonged culture time on day 7. After 2 days, the oxygen environment in all groups became stable (around 1 mg/mL) and hypoxic, but the oxygen level for the 1 mg/mL SPO/20 U catalase was still higher than the hypoxic control and improved the cell viability for 12 days.

We further encapsulated 1 mg/mL SPO/20 U catalase into fibrinogen-based hydrogels as the bioink for bioprinting applications. Results showed that this novel bioink could be bioprinted onto complex artificial muscle models for further evaluations, as shown in Figure 4a. Additionally, for the 1 mg/mL SPO/20 U catalase, significantly improved cell viability was observed from the live–dead staining images and significantly higher MTT activity detected compared to the control after culturing under the hypoxic condition for 7 days, as shown in Figure 4b,c. However, under these circumstances, cellular preservation under the hypoxic condition was not as potent as in the previous results. The reason might have been related to the significantly enhanced cell density used in the cell-laden bioink. For bioprinting applications, a high cell density at the beginning of the culture is essential for the proliferation and functionalization of tissue constructs, but here it resulted in a higher consumption of oxygen, leading to less dominant effects on the preservation of cell viability.

In addition, the cell-laden bioink containing 1 mg/mL SPO/20 U catalase further differentiated into myotubes when transferred from hypoxic to normoxic conditions, whereas the control bioink alone failed to support differentiation, as shown in Figure 5. It has been reported that hypoxia can inhibit myogenic differentiation, which corresponds with our current results for the hypoxic control [22]. However, with the 1 mg/mL SPO/20 U catalase, not only was the cell viability maintained, but the differentiation potential was protected from the hypoxic condition, which is crucial for use in tissue-engineering applications.

Overall, this study described a practical platform to evaluate the release kinetics of different oxygen-generating materials and their direct effects on cell viability under a harsh hypoxic environment. In addition, we further applied 1 mg/mL SPO/20 U catalase in fibrinogen-based bioink and observed better cell viability under hypoxic conditions, as well as improved differentiation after transferal back to normoxic conditions, which reflected the oxygen environment of the tissue-engineering application before and after vascularization, as shown in Figure 6. Even though the results are promising, this work has its limitations. First, the oxygen release rate should be further optimized for the requirements of different medical applications. Second, even though the CPO/catalase had a more appropriate oxygen release rate than the SPO/catalase for tissue-engineering applications, the CPO/catalase showed cytotoxic effects, which restrict its applications in bioprinting. Further research should be undertaken to neutralize these cytotoxic effects, which could be beneficial for long-term tissue-engineering applications with this material. Additionally, strategies to further manipulate the oxygen release of SPO/catalase should be explored to facilitate long-term tissue regeneration in bioprinting applications.

## 4. Materials and Methods

### 4.1. Materials

Sodium percarbonate (SPO) (20–30% H_2_O_2_), calcium peroxide (CPO) (75% purity), fibrinogen from bovine plasma (type I-S, 65–85% protein; ≥75% of protein was clottable), hyaluronic acid sodium salt (HA) from *Streptococcus equi*, glycerol, gelatin from porcine skin (gel strength: ~175 g Bloom, type A), thrombin from bovine plasma, aprotinin from bovine lung, and catalase from bovine liver (2000–5000 U/mg) were purchased from Sigma Aldrich (St. Louis, MO, USA).

### 4.2. Preparation of SPO and CPO

SPO and CPO powders were first cryogenically ground in a Freezer/Mill^®^ (SPEX SamplePrep LLC, Metuchen, NJ, USA) using the following protocol: 3 min of cooling in liquid nitrogen and 3 × 7 min of grinding with a magnetically driven impactor. Then, ground powders were sifted through sieves (Cole-Parmer, Vernon Hills, IL, USA) to achieve particle sizes between 46 and 72 µm prior to use.

### 4.3. Oxygen Release Kinetics

Each 50 mL centrifuge tube was filled with 30 mL of Dulbecco’s Modified Eagle Medium (DMEM) with high glucose (GE Healthcare, Chicago, IL, USA). They were maintained in an Xvivo hypoxic system (Biospherix, Parish, NY, USA) at an oxygen level of 0.1% overnight to evacuate the oxygen prior to the release study. Then, different amounts of SPO, CPO, and catalase were added to the medium and maintained in the hypoxic chamber to investigate the oxygen release kinetics of the SPO and CPO, as well as the SPO/catalase and CPO/catalase. The measurement of oxygen was performed using an Orion Versastar Pro advanced electrochemistry meter with an oxygen probe (Thermo Fisher Scientific, Waltham, MA, USA). Three samples for each composition were measured for the oxygen level, with the unit mg/mL.

### 4.4. In Vitro Cell Culture with Culture Inserts

The mouse skeletal muscle cell line C2C12 (ATCC^®^ CRL-1772™, ATCC, Manassas, VA, USA) was used to study the effects of SPO, CPO, SPO/catalase, and CPO/catalase on cell viability under the 0.1% O_2_ hypoxic condition. Twenty thousand cells were seeded in each well of several 24-well plates, along with 0.5 mL C2C12 growth medium containing DMEM, 10% FBS, and 1% penicillin, and cultured under normoxic conditions (37 °C and 5% CO_2_) for one day to allow their attachment to the well plates. This day was not counted for the culture times provided in the figures from the results.

After cellular attachment, the old medium was aspirated and 0.5 mL fresh C2C12 growth medium was used. Then, one Millicell Cell Culture Insert with an 8 µm filter (Millipore, Burlington, MA, USA) was placed on each well. Different amounts of SPO and CPO, as well as SPO/catalase and CPO/catalase, were added into another 0.5 mL C2C12 growth medium and distributed to each insert, followed by culturing in the 0.1% O_2_ hypoxic condition. The concentrations of SPO and CPO shown in the figures were based on the resulting 1 mL medium in the 24-well plates.

In addition, samples without SPO and CPO were cultured in 1 mL C2C12 growth medium under a normoxic condition and the same hypoxic condition as the normoxic and hypoxic controls, respectively. A cell-culture-insert setup was used for the culture and maintained without changing the medium.

### 4.5. In Vitro Culture of Bioprinted Muscle Construct

Our control bioink was prepared by first mixing DMEM with 10% v/v glycerol and 3 mg/mL HA at 37 °C overnight using a rotating mixer. Then, 45 mg/mL gelatin and 30 mg/mL fibrinogen were added to the solution, and it was mixed at 37 °C for another two hours using a rotating mixer. Finally, the hydrogel was sterilized by filtering it through a 0.45 µm filter prior to its use in vitro. The experimental sterilized bioink described above was further developed by adding 1 mg/mL SPO and 20 U catalase.

Bioprinting was conducted using the ITOP 3 bioprinter in the Wake Forest Institute of Regenerative Medicine (WFIRM) [23]. Prior to bioprinting, 25 million C2C12 cells were laden in 1 mL of bioink 1. The PCL was heated up to 90 °C and dispensed using a 200 µm metal nozzle at 700 psi. All bioinks were used at 15 °C and dispensed using a 200 µm Teflon nozzle at 150 psi. After the bioprinting was complete, constructs were crosslinked through immersion in 20 UI/mL thrombin solution for 1 h. Then, samples were filled with C2C12 growth medium and cultured under 0.1% O_2_ hypoxic conditions for 7 days without medium change for in vitro characterization.

After 7 days, four bioprinted samples from each group were transferred from hypoxic conditions to normoxic conditions. They were cultured, with fresh C2C12 growth medium used every other day, until day 12. Then, C2C12 differentiation medium containing DMEM/F12, 1% horse serum, and 1% penicillin was applied for these samples and maintained for 3 days. Finally, cell differentiation tests were performed for these samples using immunohistochemical analysis.

### 4.6. Live–Dead Staining

An Early Tox Live/Dead Assay Kit (Molecular Devices, San Jose, CA, USA) was used following the manufacturer-recommended protocol to observe the cell viability of C2C12 during culture. Cell-permeant red dye could bind to the DNA of all cells, while green dye could only react to dead cells without membrane integrity, giving them yellow fluorescence. Briefly, red and green dyes were diluted 1:2000 with Dulbecco’s phosphate-buffered saline (DPBS) as the working solution and then added to each well. After incubation at 37 °C and 5% CO_2_ for 20 min, fluorescence images were taken using a Spectra Max i3x with MiniMax 300 Imaging Cytometer (Molecular Devices, San Jose, CA, USA) with the excitation and emission setup provided with SoftMax Pro 7.0.2 (Molecular Devices, San Jose, CA, USA). In addition, Image J was used to quantify the cell viability based on the area of live cells, as dead cells could detach from the bottom surface.

For 3D culture using bioprinted muscle constructs, 2 µM calcein AM and 4 µM Ethidium homodimer-1 (EthD-1) solution was prepared using stock solutions from a LIVE/DEAD^®^ Viability/Cytotoxicity Kit (Thermal Fisher Scientific, Waltham, MA, USA) and added to the constructs to fully cover them. After incubation at room temperature for 30 min, an Olympus BX63 fluorescence microscope was used to take images of bioprinted muscle constructs. Green was changed to red and red was changed to green in order to attain consistency with previous live–dead images. In addition, Image J was used to quantify the cell viability based on these images. The cell viability percentage was calculated as the area of red (live cells) divided by the area of red and green (total cells) (*n* = 3).

### 4.7. MTT Assay

A 3-(4, 5-dimethylthiazol-2-yl)-2, 5-diphenyl tetrazolium bromide (MTT) assay (Sigma Aldrich, St. Louis, MO, USA) was used to evaluate the cell viability during culture. The MTT assay solution was prepared by adding 50 mg of MTT powder into 10 mL phosphate-buffered saline (PBS), followed by sterilization using 0.45 µm filters. Then, 900 µL of C2C12 growth medium and 100 µL of MTT solution were added separately to each sample. Samples were then incubated at 37 °C for 2 h. After incubation, the MTT solution was removed and 500 µL of dimethyl sulfoxide (DMSO) was added to each sample. After pipet mixing, 100 µL of the resulting solution was transferred into each well of 96-well plates and read by a Spectra Max i3x microplate reader at 570 nm. Three duplicates were tested for each composition to ensure reproducibility.

### 4.8. Immunohistochemical Analysis

Samples were fixed in 4% paraformaldehyde (PFA) in PBS solution for 10 min. Afterwards, all samples were permeabilized with 0.1% Triton X-100 in PBS solution for 10 min. Then, samples were blocked with a protein block solution (Dako, Carpinteria, CA, USA) for 30 min. The primary antibody, Myosin 4 Monoclonal Antibody (MF20) (14-6503-80, Thermo Fisher Scientific, Waltham, MA, USA) (1:300 dilution), was applied on the samples at room temperature for 1 h. After washing with PBS five times, the secondary antibody, Goat anti-Mouse IgG (H + L) Secondary Antibody Alexa Fluor 594 (A-11005, Thermo Fisher Scientific, Waltham, MA) (1:300 dilution), was applied on samples for 40 min, with aluminum foil covering to avoid light exposure. After five PBS washes, Dapi counterstain (1:1000 dilution) was applied to the samples for 5 min. After another five PBS washes, 4% PFA in PBS solution was used to fix the stain. Then, immunofluorescence images were taken using an Olympus BX63 fluorescence microscope. The myotube numbers were quantified based on these images. In addition, elongated myotubes were quantified when their aspect ratio was larger than three (*n* = 3).

### 4.9. Statistical Analysis

A one-way ANOVA statistical analysis was used and * *p*  <  0.05 was considered statistically significant.

## 5. Conclusions

In this study, the release kinetics of SPO and CPO, as well as SPO/catalase and CPO/catalase, were studied under a 0.1% O_2_ hypoxic condition. Moreover, a novel cell-culture-insert setup was used to evaluate the effects of SPO and CPO, as well as SPO/catalase and CPO/catalase, on cell viability under the same hypoxic condition. The results showed that SPO had a burst oxygen release, while CPO had a more stable oxygen release than SPO; however, both SPO and CPO showed cytotoxic effects. The presence of catalase accelerated the oxygen release of SPO and CPO, as well as improving their cell viability, but CPO remained cytotoxic with catalase. After optimization, the samples with 1 mg/mL SPO with 20 U catalase showed improved cell viability compared to the hypoxic control for 12 days. In addition, the bioink containing 1 mg/mL SPO with 20 U catalase demonstrated excellent printability, high post-printing cell viability, and unchanged differentiating potential under hypoxic conditions for 7 days. Following the current promising results, methods to further reduce the oxygen release rate of SPO should be studied in-depth to eventually overcome the challenge of vascularization in the clinical translation of bioprinting.

## Figures and Tables

**Figure 1 ijms-23-06842-f001:**
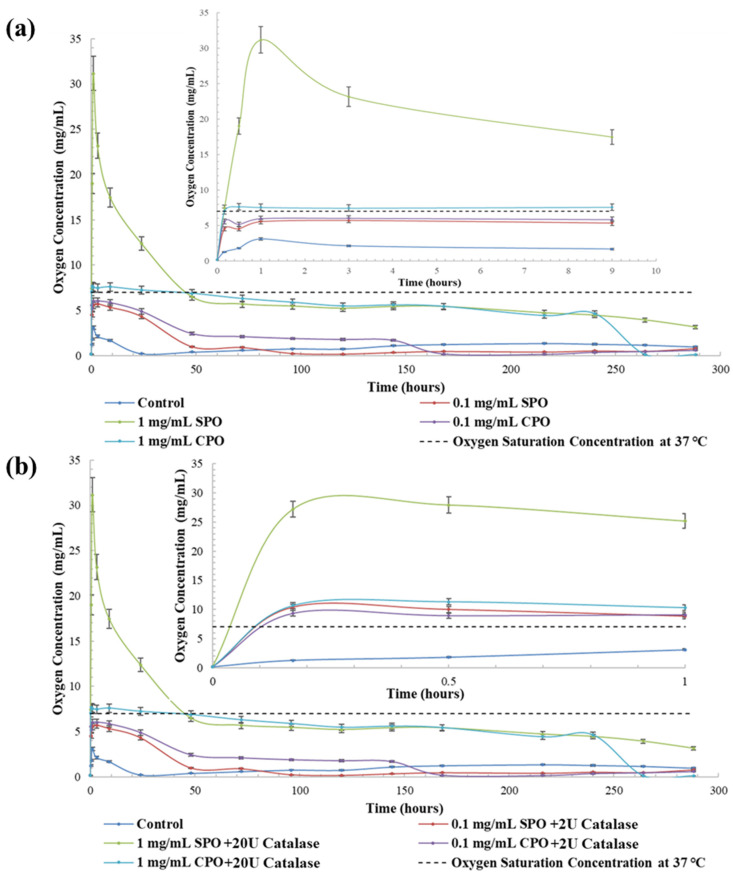
Oxygen release kinetics from (**a**) different concentrations of SPO and CPO and (**b**) different concentrations of SPO/catalase and CPO/catalase under a 0.1% O_2_ hypoxic condition. Both demonstrate that the sample with 1 mg/mL SPO in each group had the highest initial burst and maintained release profile. Inset plots depict the initial burst release curve with higher temporal resolution.

**Figure 2 ijms-23-06842-f002:**
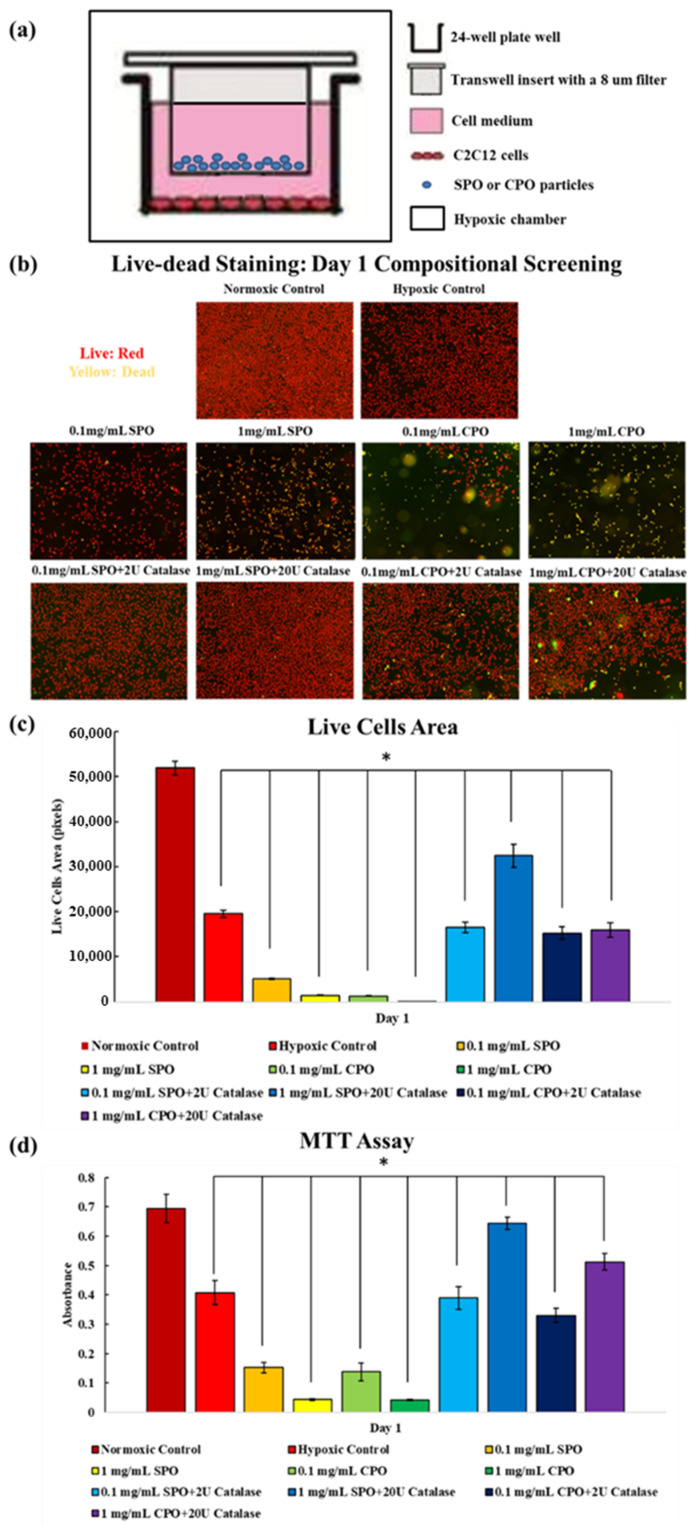
(**a**) Schematic of the novel in vitro cell-culture-insert setup. (**b**) Live–dead staining of C2C12 cells cultured for 24 h in different concentrations of SPO and CPO, as well as different concentrations of SPO/catalase and CPO/catalase. All samples were cultured under hypoxic conditions except the normoxic control. Red indicates living cells and yellow indicates dead cells. (**c**) The images were analyzed to quantify the area occupied by cells dyed red, indicating the living cell population. This measurement is compared across each treatment condition. (**d**) MTT assay results from different conditions after culturing for one day.

**Figure 3 ijms-23-06842-f003:**
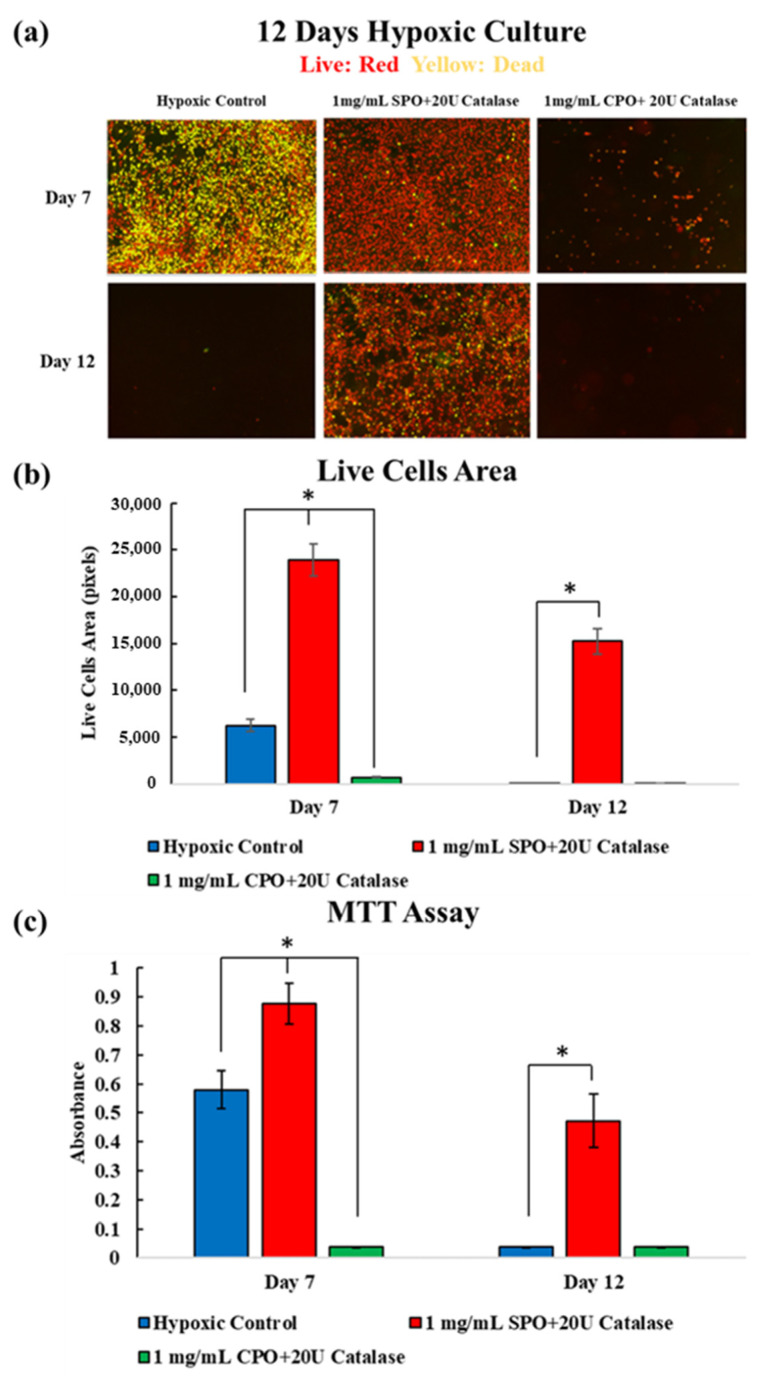
(**a**) Live–dead staining of C2C12 cells containing 1 mg/mL SPO/20 U catalase and 1 mg/mL CPO/20 U catalase after culturing under hypoxic conditions (0.1% O_2_) for 7 and 12 days. Red indicates live cells and yellow indicates dead cells. (**b**) Live–dead area of live cells calculated from the area occupied with red dyed cells indicated that 1 mg/mL SPO + 20 U catalase maintained the highest viability levels up to 12 days. (**c**) MTT assay results from the different compositions after culturing under hypoxic conditions for 7 and 12 days corroborated the live–dead staining.

**Figure 4 ijms-23-06842-f004:**
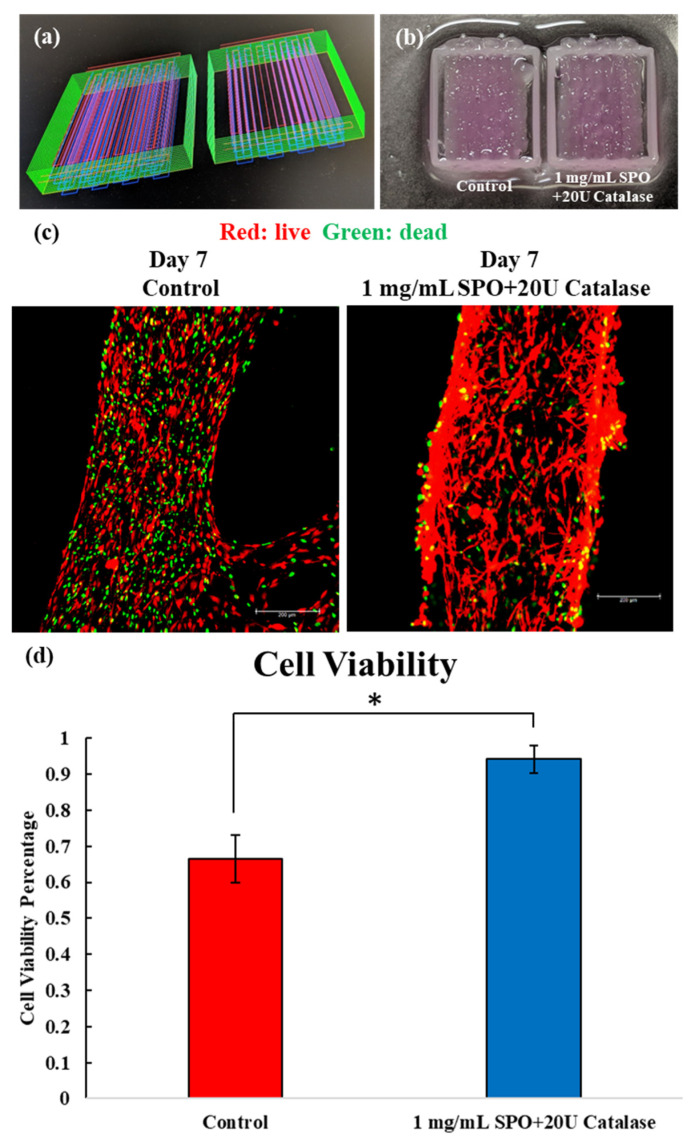
(**a**) Design of bioprinted muscle construct. (**b**) Bioprinted muscle construct using fibrinogen-based hydrogel and fibrinogen-based hydrogel with 1 mg/mL SPO + 20 U catalase. (**c**) Live–dead staining of bioprinted muscle construct after 7 days of culture under hypoxic conditions. Red indicates live cells and green indicates dead cells. (**d**) Cell viability of each composition was calculated based on the color area in the live–dead images.

**Figure 5 ijms-23-06842-f005:**
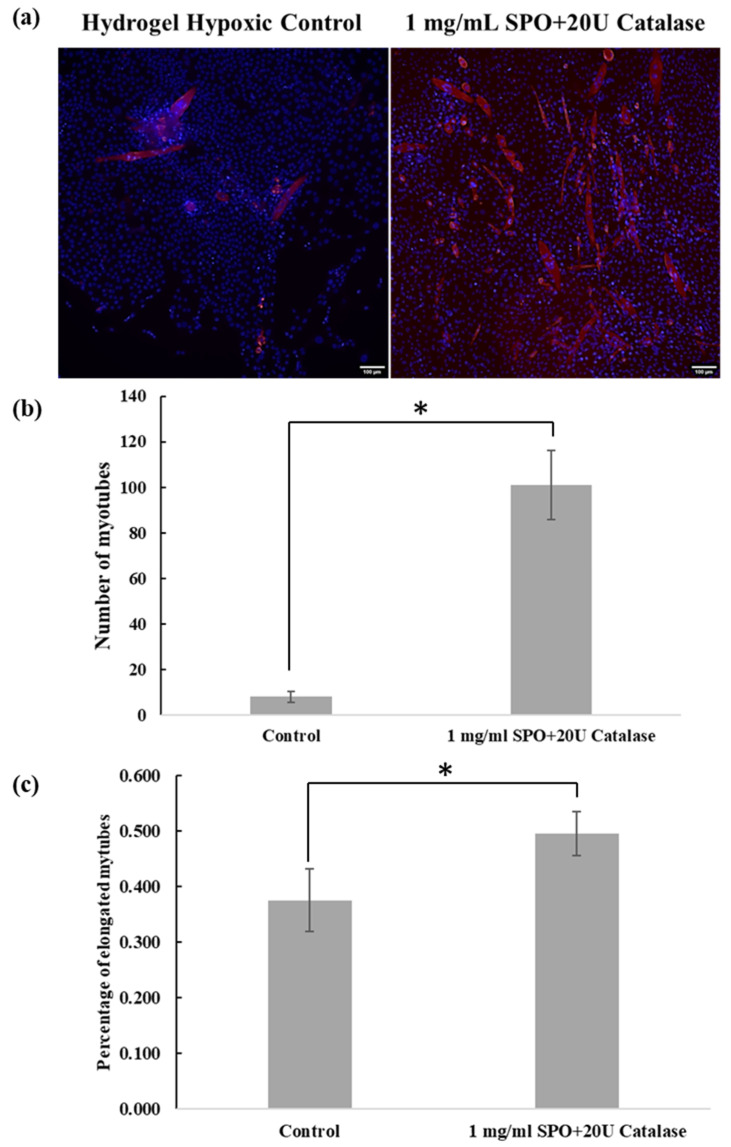
(**a**) Fluorescence images of myofibers using MF20 for bioprinted samples cultured for 8 days with control hydrogel and hydrogel with 1 mg/mL SPO/20 U catalase after transferal from hypoxic to normoxic conditions. (**b**) Quantification of total myotubes and (**c**) percentage of elongated myotubes based on fluorescence images.

**Figure 6 ijms-23-06842-f006:**
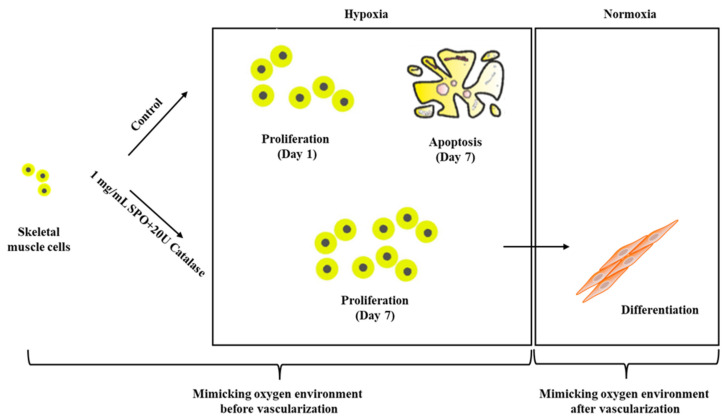
Schematic of cell preservation using 1 mg/mL SPO + 20 U catalase to mimic an oxygen environment before and after vascularization.

## Data Availability

The data presented in this study are available on request from the corresponding author. The data are not publicly available due to other ongoing investigations.

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
