# Peer review of "Release Kinetics and In Vitro Characterization of Sodium Percarbonate and Calcium Peroxide to Oxygenate Bioprinted Tissue Models"

_ijms, 2022, doi:10.3390/ijms23126842_

Round 1

Reviewer 1 Report

This study aimed to evaluate the release kinetics of sodium percarbonate (SPO) and calcium peroxide (CPO) with and without catalase in O2 condition. Then skeletal muscle cells were used to study the effects of SPO and CPO with and without catalase on in vitro cell viability. The viability of skeletal muscle cells in the presence of oxygen generation materials was also characterized through live-dead staining and MTT assay. One composition was chosen to be incorporated with hydrogel for preparing bioprinted muscle constructs followed by in vitro characterizations using live-dead staining. The differentiation of skeletal muscle cells in bioprinted constructs was also evaluated by immunocytochemical staining when transferred from hypoxic to normoxic conditions.

This contribution gives new information, it is well written and original. The topic is relevant to the field of the International Journal of Molecular Sciences journal.

The manuscript can be published in the present form.

Author Response

Response: We appreciate the positive feedback from the reviewer on the submitted article.

Reviewer 2 Report

The abstract is understandable without reading the manuscript.
The Introduction includes a description of a problem statement that conveys the important issues and provides the context for the study.
Chapter Materials and Methods contain all necessary information – Materials; Preparation of SPO and CPO;  Oxygen Release Kinetics; In Vitro Cell Culture with Culture Inserts; In Vitro Culture of Bioprinted Muscle Construct; Live-Dead Staining; MTT Assay;  Immunohistochemical Analysis and  Statistical Analysis.
The applied methods indicate the high quality of assays.
The authors presented the results of the study in an organized manner which was enhanced by precise graphic illustrations.
The discussion of data citation was good and captured the state of the art well. The conclusions are justified and based on the research presented in the paper.
I highly evaluate talent for a clear summary and interpretation of results on the background of the current literature.
The paper is written in coherent style with the use of proper terminology.
In summary, I can state that the aim of the study was achieved.
One note - Chapter: Materials and Methods should be before Chapter: Results.

Author Response

Response: For the order of manuscript sections, we followed the style recommended by the “Instructions for Authors”. If this is incorrect, we will be happy to modify it when the editor could confirm the order needed. Thank you for the positive comments on the submitted manuscript.

Reviewer 3 Report

Overall the study shows that even though both strategies are good for oxygen generation, the CPO+C strategy shows severe cell necrosis. The study is well condicted considering the superficial cell viability and oxygenation data provided and even though this may not answer muscle model research questions at all the outcome proves that not every oxygenation approach is good for a TE model.

title: the title is a bit misleading because it does not print anything relevant

fig 1 inserts are inlegible

dataset fig 3a how do dead cells disappear after timepoint 1 for hypoxia? on cpo+C shows no live cells but no yellow signal for cell death?

myotube is a strong word for this level of molecular data - can the authora furzher validate this novel muscle model more thoroughly for muscle related markers?

Author Response

Overall the study shows that even though both strategies are good for oxygen generation, the CPO+C strategy shows severe cell necrosis. The study is well conducted considering the superficial cell viability and oxygenation data provided and even though this may not answer muscle model research questions at all the outcome proves that not every oxygenation approach is good for a TE model.

Response: Thank you for the positive feedback on our submitted article.

title: the title is a bit misleading because it does not print anything relevant

Response: We appreciate the reviewer’s comments. The title has been changed to “Release Kinetics and In Vitro Characterization of Sodium Percarbonate and Calcium Peroxide to Oxygenate Bioprinted Tissue Models”

 fig 1 inserts are inlegible

Response: Thank you for pointing out this issue. Inserts in Fig 1 have been enlarged, as shown below.

 dataset fig 3a how do dead cells disappear after timepoint 1 for hypoxia? on cpo+C shows no live cells but no yellow signal for cell death?

Response: For the first time point in hypoxic and CPO+C groups, cells were dead and detached from the bottom of the well plate. Hence, no yellow fluorescence signals were observed by the microscope. To avoid this confusion, the sentence in the results section has been changed to “However, samples cultured with 1 mg/mL CPO/20U catalase showed cytotoxic effects and detached from the well plate, while demonstrated significantly lower cell viability by the same assays.”

myotube is a strong word for this level of molecular data - can the author further validate this novel muscle model more thoroughly for muscle related markers?

Response: Thank the reviewer for this very instructive comment. In this article, the primary focus is to provide a method for in-situ oxygen supply within bioprinted constructs to overcome the challenge of oxygen shortage in bioprinted constructs before their complete vascularization. As this approach has been proved to be promising in this research, we are conducting additional studies for a more thorough tissue engineered construct using this approach to address the oxygenation challenge. However, we believe that this level of characterization is outside the scope of this current manuscript. We hope the reviewer understands this situation.

Round 2

Reviewer 3 Report

The auhtors revised their manuscript accordingly.

Author Response

We appreciate the positive feedback from the reviewer on the submitted article.